# Extracellular Heat Shock Protein-90 (eHsp90): Everything You Need to Know

**DOI:** 10.3390/biom12070911

**Published:** 2022-06-29

**Authors:** Daniel Jay, Yongzhang Luo, Wei Li

**Affiliations:** 1Department of Developmental Molecular and Chemical Biology, Graduate School of Biomedical Sciences, Tufts University School of Medicine, Boston, MA 02111, USA; daniel.jay@tufts.edu; 2The National Engineering Research Centre for Protein Technology, School of Life Sciences, Tsinghua University, Beijing 100084, China; yluo@mail.tsinghua.edu.cn; 3The Department of Dermatology and the USC-Norris Comprehensive Cancer Centre, The University of Southern California Keck School of Medicine, Los Angeles, CA 90033, USA

**Keywords:** extracellular Hsp90, stress, mechanism of action, wound healing and cancer

## Abstract

“Extracellular” Heat Shock Protein-90 (Hsp90) was initially reported in the 1970s but was not formally recognized until 2008 at the 4th International Conference on The Hsp90 Chaperone Machine (Monastery Seeon, Germany). Studies presented under the topic of “extracellular Hsp90 (eHsp90)” at the conference provided direct evidence for eHsp90’s involvement in cancer invasion and skin wound healing. Over the past 15 years, studies have focused on the secretion, action, biological function, therapeutic targeting, preclinical evaluations, and clinical utility of eHsp90 using wound healing, tissue fibrosis, and tumour models both in vitro and in vivo. eHsp90 has emerged as a critical stress-responding molecule targeting each of the pathophysiological conditions. Despite the studies, our current understanding of several fundamental questions remains little beyond speculation. Does eHsp90 indeed originate from purposeful live cell secretion or rather from accidental dead cell leakage? Why did evolution create an intracellular chaperone that also functions as a secreted factor with reported extracellular duties that might be (easily) fulfilled by conventional secreted molecules? Is eHsp90 a safer and more optimal drug target than intracellular Hsp90 chaperone? In this review, we summarize how much we have learned about eHsp90, provide our conceptual views of the findings, and make recommendations on the future studies of eHsp90 for clinical relevance.

## 1. Introduction

For decades, the Heat Shock Protein-90 (Hsp90) family proteins have been recognized as ATP binding-dependent molecular chaperones inside almost all types of cells throughout evolution. This understanding has served as an indisputable foundation for both laboratory research and cancer clinical trials targeting the intracellular function of the Hsp90 family proteins [1,2,3,4,5,6]. Meanwhile, a cell-surface form of Hsp90-related molecule was reported as early as the late 1970s with several publications that appeared to challenge the definition of Hsp90 as an exclusively intracellular chaperone. The question was first raised in the 1990s by Csermely and colleagues, who stated that “the major cellular function of Hsp90 is probably not its chaperone behaviour, but its dynamic participation in the organization and maintenance of the cytoarchitecture” [7], although the exact nature of the dynamic participation was not further elaborated. Throughout the following decade, however, few in the Hsp90 field credited the possible existence of a non-chaperone form of Hsp90 and regarded the reported extracellular or secreted Hsp90 as artifacts, such as leakage by a small number of dying cells in culture. A breakthrough emerged in the 2000s when several laboratories independently demonstrated a critical role for secreted Hsp90 in various pathophysiological processes such as cancer cell invasion and wound healing. Increasing lines of evidence are raising the possibility that cell surface-bound, exosome-anchored, or simply free-secreted Hsp90 may serve as safer and more effective therapeutic targets than their intracellular counterparts in cancer and other inflammatory human disorders. This is especially relevant considering that targeting the intracellular ATP-dependent chaperone function of Hsp90 has encountered setbacks in clinical trials. Several generations of small molecule inhibitors have entered numerous cancer clinical trials since 1999, but to date none have received FDA approval [5,6]. In this review article, we provide a comprehensive walk-through of the discovery, characterization, mechanism of action, and evaluation by animal models and human patients of what is now collectively referred to as “extracellular Hsp90” (eHsp90). More importantly, we offer our answers for the fundamental question of why eHsp90 is chosen by evolution for duties that cannot be served by other conventional extracellular factors. 

## 2. History of eHsp90 Discovery

In the late 1970s, several laboratories independently reported a glucose-regulated 90-kDa protein both on the surface and in the conditioned medium of tumour virus-infected mouse and human fibroblast cells [8,9,10,11,12]. In 1983, Hughes et al. provided direct evidence that Hsp90 protein is located on the external surface of macrophage and mouse embryo 3T3 cells [13]. Srivastava et al. then reported a membrane-associated 96-kDa protein in chemically induced sarcoma cells [14] and Ullrich et al. showed that a Hsp90-related protein was detected on the external surface of both Meth A tumour and NIH3T3 cells using antibody binding to the cells at 4 °C that prevented membrane internalization [15]. A follow-up study by Thangue and Latchman showed cell surface accumulation of Hsp90 in HSV-infected cells [16]. While the findings of these studies were intriguing, they may have not resonated at the time. The observations of extracellular Hsp90 in these studies were thought to be due to intracellular Hsp90 being released non-specifically from dead or dying cells, and there was little preclinical or clinical relevance available. In 1992, Erkeller Yuksel et al. reported that the external surface expression of Hsp90 was a feature of about 20% of the patients with systemic lupus erythematosus (SLE) and it correlated with the severeness of the disease [17]. After the finding, Thomiadou and Patsavoudi reported a 94-kDa “neuron-specific cell surface antigen” recognized by the monoclonal antibody 4C5 [18]. The 94-kDa protein was later identified as an Hsp90-related protein by the same group using mass spectrometry [19]. The take-home message of these studies was the association of the surface or secreted Hsp90 with inflammatory diseases and tissue development. The findings of these earlier studies were, however, largely overlooked by the Hsp90 community due to lack of evidence for active secretion and the undefined function of surface-bound or secreted forms of Hsp90.

In the early 2000s, two laboratories, which had never studied Hsp90 before, were independently searching for secreted proteins that support two distinct and mechanistically related pathophysiological processes, tumour cell invasion and skin wound healing. In 2004, Jay’s group at Tufts University first reported the identification of a secreted protein from the conditioned medium of a fibrosarcoma cell line, HT-1080, and showed that the secreted protein promoted tumour cell invasion in vitro by activating the matrix metalloproteinase 2 (MMP2) [20]. In 2007, Li’s laboratory at the University of Southern California reported the purification of a secreted protein from the conditioned medium of hypoxia-stressed primary human dermal fibroblasts and keratinocytes through chromatography and showed that this secreted protein strongly stimulated skin cell migration in vitro and promoted wound healing in mice [21,22]. The common protein involved in tumour cell invasion in vitro and wound healing in vivo was identified as the secreted form of Hsp90α. Additional publications on secreted Hsp90α have since emerged and are beginning to receive attention. To provide a common terminology that covers the meanings of “cell surface-bound”, “cell-released”, “cell-secreted” Hsp90α, Isaacs and colleagues recommended “eHsp90” for extracellular Hsp90 which has since become widely accepted by the Hsp90 community [23]. Over the past 20 years, there have been two dozen excellent review articles on eHsp90, especially eHsp90α [24,25,26,27,28,29,30,31,32,33,34,35,36,37,38,39,40,41], which we use as the stepping-stones for construction of this article.

## 3. eHsp90α vs. eHsp90β: Who Calls the Shots and Why?

In comparison to over 70 reports on eHsp90α as of May 2022, several studies in the past have also reported the presence of either eHsp90β alone or the eHsp90α and eHsp90β proteins together in the conditioned medium of various cell types [42,43]. This leads to the question of whether eHsp90β also has extracellular functions. If one only considers the functionality of a protein in its purified form as the ultimate evidence, the answer is negative. Cheng et al. showed that human recombinant Hsp90α (hrHsp90α), but not hrHsp90β, stimulated human keratinocyte migration [22]. Jayaprakash et al. showed that hrHsp90α, but not hrHsp90β, promoted wound healing in pigs [44]. Zou and colleagues demonstrated that the intravenous injection of hrHsp90α, but not hrHsp90β, protein strongly promoted tumour formation and lung metastasis in mice [45].

The human Hsp90α and Hsp90β proteins differ by a total of 100 amino acid residues along their respective 732- (Hsp90α) and 724- (Hsp90β) amino acid sequences, including 58 conservative and 42 non-conservative amino acid substitutions, in addition to 12 amino acid deletions in Hsp90β. The highest variations between Hsp90α and Hsp90β occur within the linker region (LR) and a part of the middle domain (M), with only 61% amino acid identity, the location where Li’s group identified the functional “F-5” fragment of eHsp90α that promotes wound healing [46,47]. Mouse genetic studies also showed distinct and non-compensating roles for Hsp90α and Hsp90β during development. Voss et al. reported that Hsp90β gene knockout causes a defect in placental labyrinth formation, resulting in mouse embryonic lethality on E10.5 [48]. In contrast, mice with either chaperone-defective mutations in Hsp90α [49,50] or complete Hsp90α knockout [51] showed indistinguishable difference in their phenotypes from their wild-type counterparts. The straightforward interpretations were that (1) Hsp90β is more critical than Hsp90α during mouse development and (2) Hsp90α is not required under homeostasis.

## 4. Is eHsp90 Secreted by Living Cells on Purpose or Leaked by Dead Cells by Accident?

While it is technically difficult to prove that eHsp90, specifically eHsp90α, does not result from the leakage of intracellular Hsp90 from a small number of dead cells in culture, several lines of evidence strongly support that eHsp90α is actively secreted. Studies showed that the quantity of eHsp90α was less or undetectable from secreted molecules of normal cells under physiological conditions in vitro, i.e., in serum-containing and pH-balanced medium under normoxia at 37 °C. In comparison, several fold higher eHsp90α proteins became detectable from the conditioned medium of the same cells under a variety of medically-defined stress signals including reactive oxygen species (ROS), heat, hypoxia, gamma-irradiation, UV, and tissue injury [28,52,53]. In contrast, many tumour cells constitutively secrete eHsp90α due to intrinsic oncogenes, such as overexpressing HIF-1α [21], or mutant forms of tumour suppression genes including p53 [42,54]. Eustace et al. showed only Hsp90α and not Hsp90β in the conditioned medium of tumour cells, suggesting a specific secretion of Hsp90α rather than the non-specific release of both forms from dead cells [20]. Cheng and colleagues showed that while both TGFα and EGF bind and signal through EGFR and both promote cell survival and cell growth, only TGFα stimulates the Hsp90α translocation to plasma membrane and secretion to the extracellular environment by primary human keratinocytes, the most critical cell type for skin wound healing [22]. Finally, the stress-induced secretion of Hsp90α was further substantiated by an in vivo observation that skin injury caused up to a 10-fold increase in eHsp90α deposition into the wound bed in a time-dependent fashion. As shown in Figure 1, skin injury in pigs causes an accumulatively increased deposition of eHsp90α into the wound bed in a time-dependent fashion. Since the location with increased anti-Hsp90α antibody staining includes areas in the skin dermis that does not have the continued presence of cells, the massive staining cannot be explained by increased intracellular Hsp90α [55]. This finding provides the first in vivo evidence of biological stress-induced Hsp90α secretion.

So far, the studies on mechanism(s) of eHsp90α secretion have raised more questions than answers. Two laboratories reported that the secretion can be regulated by either phosphorylation or the C-terminal amino acid EEVD motif of the Hsp90α protein [56,57]. Luo’s laboratory further identified a critical role for Rab coupling protein (RCP) in mutp53-induced Hsp90α secretion [54]. Several studies suggested that eHsp90α is secreted via exosomes, based on the observation that DMA (Dimethyl amiloride), an inhibitor of the exosome secretion pathway, blocks Hsp90α secretion both in HIF-1-overexpressing tumour cells and TGFα-stimulated human keratinocytes cells, where Hsp90α was associated with isolated exosome fractions. Therefore, eHsp90α is secreted via the non-classical exosome trafficking pathway [58,59,60,61]. Guo and colleagues further identified the proline-rich Akt substrate of 40 kDa (PRAS40) as the unique downstream effector that mediates TGFα-stimulated Hsp90α secretion via exosomes [62]. Tang and colleagues have recently made an interesting observation that eHsp90α is located at the external surface of tumour cell-secreted exosomes [63]. However, recent observations suggest that approximately 90% of both normal and tumour cell- secreted Hsp90α is not associated with secreted exosomes isolated by ultracentrifugation (C. Cheng, X, Tang and W. Li, unpublished; A. Bernstein and D. Jay, unpublished). Taken together, eHsp90α is a secreted protein by cells under either internal or external stress. This general understanding is depicted in Figure 2, where eHsp90α promotes tissue repair under physiological conditions or promotes tumorigenesis under pathological conditions, defined as a “double-edged sword” by Hence and colleagues [30].

## 5. Two Main Biological Functions of eHsp90α

### 5.1. Promoting Cell Survival under Ischemic Stress

Shortly after tissue injury, the broken blood vessels clot and cells in the injured tissue encounter an ischemic (paucity of nutrient and oxygen) environment. The immediate challenge the cells face is survival, at least temporarily, by adapting a self-supporting mechanism. Similarly, when tumour cells invade surrounding tissues too quickly and temporarily outstrip the nearest blood vessel for 150 mirom or more, they similarly face the stress of ischemia [64]. Under these conditions, the tumour cells must find an autocrine cycle to survive without the help from blood vessels. Bhatia and colleagues showed that topical application of hrHsp90α to burn wounds in pigs prevented heat-induced skin cell apoptosis around the hypoxic wound bed [65]. Dong and colleagues demonstrated that eHsp90α protected tumour cells from hypoxia-triggered apoptosis, whereas neutralizing eHsp90α function with a monoclonal antibody enhanced hypoxia-induced tumour cell apoptosis [66]. Gao and colleagues reported that extracellular supplementation with hrHsp90α (10 μM or ~1 mg/mL) protein promoted rat bone mesenchymal stem cell (MSC) survival and prevented cell apoptosis under ischemic conditions by activating the Akt and ERK kinases [67]. Cheng et al. showed that 10 μg/mL hrHsp90α protein stimulated the maximum migration of primary human skin cells [22] and Dong et al. reported that a similar dosage of hrHsp90α prevented the death of Hsp90α-KO MDA-MB-231 cells under hypoxia [66]. Nonetheless, the conformation and composition of the eHsp90α under the above circumstances is less clear in comparison to its intracellular counterpart.

### 5.2. Promoting Cell Motility (Not Growth) during Tissue Repair and Tumour Invasion

The initial indication that eHsp90α regulates cell migration was reported in the 1990s by Patsvoudi’s group, who showed that a monoclonal antibody against a mouse granule cell surface antigen called 4C5 inhibited the cell migration during cerebellar development [68,69]. The same group later confirmed by immunoprecipitation followed by mass spec that the 4C5 antigen is related to Hsp90 protein [19]. During the same period, Jay’s group showed that eHsp90α from the conditioned medium of tumour cells was required for tumour cell invasion via activation of MMP2 in vitro [20]. The direct evidence that eHsp90α protein alone acts as a *bona fide* pro-motility factor came from Li’s group that demonstrated hrHsp90α, but not hrHsp90β, stimulated primary human dermal fibroblasts and keratinocyte migration in the total absence of serum factors. Moreover, the pro-motility effect of hrHsp90α could reach approximately 60% of the total pro-motility of 10% FBS-containing medium. Under similar conditions, however, hrHsp90α showed little mitogenic effect on cell growth. More surprisingly, both the wild type and ATPase-defect mutant proteins of Hsp90α bind the cell surface receptor LRP-1 (low-density lipoprotein receptor-related protein 1) and had compatible prom-motility effects on the same cells [21,22].

## 6. Mechanisms of Action by eHsp90α

By and large, there have been two major parallel mechanisms of action proposed for eHsp90α [28]. The central debate is whether eHsp90α still acts as an ATP-dependent chaperone outside the cell or alternatively acts as a previously unrecognized signalling molecule no longer dependent on ATP hydrolysis. Eustace and colleagues tested DMAG-N-oxide, a cell membrane-impermeable geldanamycin/17-AAG-derived inhibitor that targets the ATPase activity of Hsp90, and showed that it inhibits tumour cell invasion [20]. Similarly, Tsutsumi and colleagues showed that the DMAG-N-oxide inhibitor reduced the invasion of several cancer cell lines in vitro and lung colonization by B16 melanoma cells in mice [70]. Furthermore, Sims et al. showed that blocking ATPase using ATP-gamma S actually increased the ability of hrHsp90α to activate MMP2 in vitro [71]. In particular, a recent elegant study from Bourboulia‘s group showed that TIMP2 and AHA1 act as a molecular switch for eHsp90α that determines the inhibition or activation of the eHsp90α client protein MMP2 [72]. Song and colleagues showed that Hsp90α, but not Hsp90β, stabilized MMP2 and protected it from degradation in tumour cells in an ATP-independent manner and was mediated by the middle domain of Hsp90α binding to the C-terminal hemopexin domain of MMP2 [73]. Taken together, these studies suggest that the N-terminal ATP-binding domain and the intrinsic ATPase of Hsp90α remain essential for eHsp90α function outside of the cells. Results of other studies from different laboratories also supported the “eHsp90α chaperone mechanism” via their extracellular client proteins, most noticeably MMP2, MMP9, and TLR, just to mention a few. To avoid redundance in this special issue, we refer readers to two excellent review articles, a prior one by Wong and Jay [32] and the current one in this special issue by Bourboulia and colleagues for more detailed analysis of this mechanism.

On the other hand, the ATPase-independent mechanism has largely focused on the so-called “eHsp90α > LRP-1” signalling pathway [28]. Li’s laboratory utilized both deletion and site-directed mutagenesis to narrow down the essential epitope along the 732-amino acid human eHsp90α for supporting the pro-survival, pro-motility, and pro-invasion activity of eHsp90α in vitro and in vivo. First, Cheng and colleagues reported that the ATPase-defective mutants, Hsp90α-E47A (~50% ATPase activity), Hsp90α-E47D (ATPase-defect), and Hsp90α-D93N (ATPase-defect), showed an indistinguishable degree of pro-motility activity from the Hsp90α-wt protein on primary human skin cells in vitro [22]. Second, they narrowed down the pro-motility activity to a 115-amino acid fragment called F-5 (aa-236 to aa-350) between the LR (linker region) and the M (middle domain of human) Hsp90α, as previously mentioned. They demonstrated that the F-5 peptide alone promoted skin cell migration in vitro and wound healing in vivo as effectively as the full-length Hsp90α-wt [46]. Third, they illustrated the so-called “eHsp90α > LRP-1” signalling pathway as: (1) the subdomain II in the extracellular part of the low-density lipoprotein receptor-related protein-1 (LRP-1) that receives the eHsp90α signal; (2) the NPVY, but not NPTY, motif in the cytoplamic tail of LRP-1 that connects the eHsp90α signalling to the serine-473, but not threonine-308, phosphorylation in Akt kinases and (3) activated Akt1 ang Akt2 trigger cell migration [47]. Finally, within the F-5 fragment, Zou and colleagues identified a dual lysine motif (Lys-270/Lys-277) that are evolutionarily conserved in all members of the Hsp90α subfamily but absent in all Hsp90β subfamily members. Mutations at the lysine residues eliminated Hsp90α’s ability to promote cell migration in vitro and tumour formation in vivo. Substitutions of the two different amino acids at the corresponding sites in Hsp90β granted Hsp90β with pro-motility activity like Hsp90α [45]. These authors presented an illustration of the F-5 fragment and the dual lysine motif locations in a schematic monomer structure of Hsp90α, as shown in Figure 3, which shows a potential target in eHsp90α for therapeutics. These findings suggest that the N-terminal ATPase domain and the C-terminal dimer-forming and co-factor-binding domain are dispensable for eHsp90α function. More interestingly, Gopal showed a novel crosstalk mechanism involving the eHsp90α-LRP1 dependent regulation of EphA2 function, in which the eHsp90α-LRP1 signalling axis regulates AKT signalling and EphA2 activation during glioblastoma cell invasion [23]. In addition, Tian et al. showed that clusterin served as an eHsp90α modulator to synergistically promote EMT (epithelial-to-mesenchymal transition) and tumour metastasis via LRP1 [74]. Besides binding to MMP2 and LRP-1, Garcia et al. reported that eHsp90α binds to the type I TGFβ receptor to stimulate collagen synthesis, which provides pavement for cell attachment and migration [75]. Nonetheless, the chaperone-dependent mechanism, such as the activation of MMP2, and the chaperone-independent mechanism, such as F-5 binding the LRP-1 receptor, do not necessarily have to be mutually exclusive, as schematically depicted in Figure 4, which may represent two parallel mechanisms of action by eHsp90α. The selectivity and specificity of these two pathways under various pathophysiological conditions remain to be further studied.

## 7. Preclinical Studies of eHsp90α

eHsp90α has been studied in several human disease models in animals, including cancer, wound healing, idiopathic pulmonary fibrosis (IPF) and wasting syndrome (WS). The implication of eHsp90α in blood circulation supporting tumour metastasis in a number of animal models is especially encouraging considering the long-term and heavy emphasis of Hsp90 on cancer.

### 7.1. Wound Healing

When tissue is injured and the broken blood vessels clotted, all of the cells surrounding the wound bed face ischemic stress, as previously described. Bhatia and colleagues made full-thickness skin wounds in pigs, biopsied the wounds over time, and immunostained the tissue samples with an anti-Hsp90α antibody. They found a massive, time-dependent increase in the antibody staining in both the epidermis and dermis [56] (see Figure 1). Song and Luo reported that eHsp90α localized on blood vessels in the granulation tissue of wounded skin and promoted angiogenesis during wound healing in mice [76]. A series of studies from Li’s group demonstrated that topical application of hrHsp90α, but not hrHsp90β, strongly promoted closure of trauma (excision), burn, and diabetic wounds in mice and pigs. In reserves, topically administered antibodies against eHsp90α blocks wound closure [21,44,55,65,77]. Bhatia and colleagues carried out a clever study by taking advantage of Hsp90α transgenic mice where the Hsp90α’s intracellular chaperone function is nullified but the truncated Hsp90α protein still contains the entire F-5 region. They found that these mice heal skin wounds as efficiently as their wild-type counterparts, indicating that the chaperone function of Hsp90α is dispensable. However, topical application of mAb 1G6D7 against eHsp90α inhibited the wound healing, suggesting an essential role for eHsp90α instead [55]. As previous mentioned, the wound healing-promoting effect of the full-length eHsp90α is entirely replicable by the F-5 fragment [46], which is currently undergoing clinical trials for the treatment of diabetic foot ulcers.

### 7.2. Tissue Fibrosis

While Hsp90α enhances wound healing, excess eHsp90α in the injured lung may do more harm than good [78]. Pulmonary fibrosis is characterized by overactivated lung fibroblasts and massive collagen deposition by the cells at the injured site. Using BLM-induced pulmonary injury and the fibrosis mouse model, which represents failed wound healing, Dong and colleagues showed that mAb 1G6-D7, a monoclonal antibody against eHsp90α, inhibited eHsp90α function and significantly protected against BLM-induced pulmonary fibrosis by ameliorating fibroblast overactivation and ECM production [79]. The same group proposed a possible mechanism by which eHsp90α links the ER stress to the PI-3K-Akt pathway [80]. Ballaye and colleagues reported the significant increase of both eHsp90α and eHsp90β in the circulation of patients with idiopathic pulmonary fibrosis (IPF), and the higher levels correlated with disease severity. They found that eHsp90α signalled through LRP-1 to promote myofibroblast differentiation and persistence in a rat ex vivo model [81]. Together, the above studies argue that the specific inhibition of eHsp90α is a promising therapeutic strategy to reduce pro-fibrotic signalling in IPF.

### 7.3. Wasting Syndrome

Wasting syndrome (WS) refers to the unwanted weight loss of more than 10 percent of a person’s body weight, with diarrhea, weakness, and fever that can last up to 30 days. WS is often a sign of disease, such as cancer, AIDS, heart failure, or advanced chronic obstructive pulmonary disease. “Cachexia”, characterized by muscle wasting, is a major contributor to cancer-related mortality. A recent study by Zhang et al. reported elevated serum Hsp70 and Hsp90 in Lewis lung carcinoma (LLC)-bearing mice. The tumour-released and exosome-bound eHsp90 and eHsp70 were both necessary and sufficient to induce muscle wasting in a syngeneic tumour mouse model [82]. These studies may suggest clinical value in inhibiting eHsp90 for WS.

### 7.4. Tumorigenesis

Given the specific supporting role of eHsp90α in cancer, and the failure of many clinical trials using pan- inhibitors targeting all intracellular Hsp90 chaperone members, several groups have reported on the benefit of selectively inhibiting eHsp90α for reducing tumour metastasis and improving patient survival. Stellas et al. reported that intraperitoneal injection with monoclonal antibody (100–200 μg per mouse daily) against 4C5 antigen (cell surface eHsp90α) into C57BL/6 mice 24 h following tail vein injection with B16-F10 melanoma cells reduced tumour lung colonization and improved the survival of the mice in reference to placebo-treated mice [83]. These authors reported a similar finding using a human breast cancer xenograft model and showed that the antibody disrupted interactions of eHsp90 with MMP2 and MMP9 [84]. Tsutsumi and colleagues showed that DMAG-N-oxide (a membrane-impermeable version of 17-DMAG inhibitor) blocked lung colonization by B16 melanoma cells in nude mice [71]. Results of the study suggest that the N’-terminal ATP-binding of Hsp90α is still required for eHsp90α function. Using a different anti-Hsp90α monoclonal antibody, Song and colleagues showed the dose-dependent inhibition of tumour growth and angiogenesis of B10-F10 cells in nude mice [74]. Using orthotopic breast cancer mouse models, Hou and colleagues showed that the injection of hrHsp90α protein increased primary tumour lymphatic vessel density and sentinel lymph node metastasis. In reverse, injection of another independent anti-Hsp90α neutralizing antibody reduced 70% of lymphatic vessel density and 90% of sentinel lymph node metastasis [85]. Using the well-known human triple negative breast cancer cell line, MDA-MB-231, xenograft mouse model, two laboratories independently reported a critical role for eHsp90α in tumour growth and lung metastasis. Stivarou and colleagues showed that injection with an antibody against 4C5 antigen (Hsp90) inhibited both *de novo* tumour growth and growth of already established mammary tumours [86]. Zou and colleagues demonstrated that injection with hrHsp90α, but not hrHsp90β, protein rescued the tumorigenesis of Hsp90α-knockout MDA-MB-231 cells in nude mice. More interestingly, the authors showed that the ATPase-defective Hsp90α (Hsp90α-D93N) protein showed exactly the same effect as the wild type Hsp90α on tumour formation and lung metastasis. In reverse, injection with the monoclonal antibody mAb1G6-D7, not only blocked de novo tumour formation and lung metastasis, but also significantly reduced (~35%) the continued growth of already formed tumours [45]. Consistently, Secli el al recently reported that “Morgana”, a co-chaperone of eHsp90α, induced cancer cell migration through TLR2, TLR4, and LRP1. A monoclonal antibody targeting Morgana inhibited mouse breast cancer cells, EO771, from metastasizing to the lung in C57BL/6 mice [87]. Milani and colleagues established mouse models with human acute lymphoblastic leukemia (ALL) and showed that the background plasma level of eHsp90α was below 1ng/mL blood in healthy mice, whereas the plasma level of eHsp90α was elevated into the 100–150 ng/mL range within two months in a fashion that closely correlated with the increased percentage of hCD45+ cells, a monitoring marker of ALL, in the blood, bone marrow, liver, and spleen of the animals [88]. A recent study by Luo’s group showed that PKM2 (pyruvate kinase M2)-like eHsp90α is secreted by lung cancer cells and detected in blood samples of human cancer patients. The injection of mouse recombinant PKM2 protein into blood circulation promoted tumour metastasis to the lung via binding to integrin β1 [89]. Since PKM2 is associated with Hsp90α inside cells [90], it is possible that the secreted PKM2 is in complex with eHsp90α, which remains to be experimentally confirmed.

## 8. Clinical Studies of eHsp90α in Patients with Cancer and Inflammatory Disorders

Since 2008, close to two dozen clinical studies have compared the eHsp90α levels in blood circulation between healthy humans and patients with various types of cancers and other inflammatory diseases. The cancers from the patients include all of the NCI (National Cancer Institute, USA)-listed major human cancers. Due to space limitations, we are unable to describe each of the individual studies and their findings in detail. Rather, we chose to summarize the common findings of these studies, i.e., elevated plasma eHsp90α in circulation, in Table 1. While the exact amount of plasma/serum eHsp90α markedly vary (from pg/mL to mg/mL) among different reports (though they all used ELISA-based detection methods), a majority of the studies showed a statistically significant increase in cancer patients compared to normal patients in a range of sub μg/mL. More intriguingly, the higher levels of plasma eHsp90α closely correlated with the later stages of the diseases, such as metastasized tumours. These studies raise the possibility of utilizing plasma eHsp90α as a new serum marker for cancer detection and therapeutic targeting, as well as for other chronic inflammatory diseases in humans.

Taking tumorigenesis as an example, the recognized five steps of tumour development include gene mutations, hyperplasia, dysplasia, primary tumour formation, and tumour metastasis [119,120,121,122]. The vast majority of the United States Food and Drug Administration (FDA)-approved oncology drugs (>1000 by the end of 2020) target primary tumours, even though cancer patients die predominantly from metastasis [123]. These drugs extend patients’ survival for variable periods of time, but many lose efficacy shortly after several months of treatment due to new mutations generated in the tumours. On the other hand, tumour metastasis begins with local expansion and invasion of the tumour at the primary organ driven by oncogenic signals with tumour microenvironmental assistance. Tumour cells migrate away from their origin and infiltrate into new surrounding tissues in which the tumour cells intravasate into the nearest blood circulation or the lymphatic system. After entry into the circulation, the tumour cells become known as circulating tumour cells (CTCs). Continued distal metastasis requires the tumour cells to survive and disseminate via the blood circulation, so-called hematogenous metastasis. Only a small number of CTCs successfully extravasate by crossing the endothelial barrier, leaving the circulation, and entering a distant organ. Thus, identification of a plasma factor that provides critical assistance for CDC to achieve the ultimate success of metastasis could lead to the sought-after target for next generation of anti-tumour therapeutics. If elevated plasma eHsp90α in cancer patients proves to promote CTC survival and dissemination through blood circulation during metastasis, interruption of the plasma eHsp90α function by antibodies that target the F-5 region of eHsp90α, as schematically proposed in Figure 5, would be an attractive approach to slow down tumour metastasis and buy time for patients to eliminate the primary tumours via surgery and the currently available therapies. For the next few years, the potential importance of the plasma eHsp90α reported in human cancer patients must be carefully studied by engineering the pathological plasma eHsp90α levels in Hsp90α-knockout animal models.

## 9. Is eHsp90α a More Effective and Safer Drug Target than Intracellular Hsp90?

As mentioned at the beginning of this article, over the past two decades, intracellular Hsp90 chaperones (Hsp90α, Hsp90β, and possibly other related chaperones) have been targeted by at least 18 small molecule inhibitors binding to the N-terminal ATP/ADP binding site of the proteins in more than 60 cancer clinical trials [4,5,6]. To date none has received FDA approval for clinical treatment of human cancers due to various speculative reasons [124]. A recent study raised a serious and previously overlooked concern that there might be a complete lack of a druggable window between tumour and normal tissues for ATP-binding inhibitors. Tang and colleagues showed a wide range of Hsp90 expression in different host organs which further exhibited a wide range of toxicity to an ATP-binding inhibitor and heterogenous responses against the conversional theory to the same ATP-binding inhibitor among different tumour cells. These findings could seriously complicate patient and biomarker selections, toxicity readout, and efficacy of the drug candidates for clinical trials [125]. In contrast to the essential role of the intracellular Hsp90, especially Hsp90β for cell and organ homeostasis, the requirement of eHsp90α for life has not been reported. Instead, only when tissue homeostasis is broken, such as during wound healing or disease occurrence such as tumour growth, does eHsp90α then come into the picture. To support this notion, Bhatia and colleagues showed that selectively blocking eHsp90α by antibodies delayed wound healing [55]. Similarly, CRISPR-knockout of the Hsp90α gene selectively eliminated the ability of the MDA-MB-231 tumour cells to invade a Matrigel barrier and form tumours in mice. More remarkably, the defective tumorigenicity of Hsp90α-KO tumour cells could be fully rescued by extracellular supplementation with hrHsp90α proteins in an ATPase-independent fashion [45]. Therefore, in theory, drugs targeting eHsp90 should achieve higher efficacy and pose minimum toxicity to patients. A schematic representation of this simplified thought is depicted in Figure 6.

## 10. Why Is eHsp90α Co-Opted for Extracellular Duties?

Our current understanding of this fundamental question remains little beyond speculation. An entry point to understand the question is the fact that Hsp90 maintains an unusually high expression level in almost all cell types. Although the statement of “1–2% Hsp90 of total cellular proteins” has been used for decades, this number did not come from direct experimental measurements, but rather from estimations. The first quantitation of the cellular Hsp90 protein was completed by Sahu and colleagues in 2012. Using classical biochemical techniques, these authors demonstrated that Hsp90 accounted for 2–3% of the total cellular proteins among four normal cell lines and 3–7% of the total cellular proteins among four cancer cell lines tested [126]. More surprisingly, a recent study involving 12 (eight tumour and four normal) cell lines reported a much greater variation in the total cellular Hsp90 (α and β) expression, a range of 1.7% to 9% among non-cancer cell lines and different mouse organs and a range of 3 to 7% among the tumour cell lines [125]. If we take the general assumption that a given type of human cell expresses 1/3 of its total 30,000 protein-coding genes, the percentage of the Hsp90 expression is at least several hundred times higher than the rest of the 9999 cellular gene products. The question is why a particular gene product must be given such a spatial privilege. Evolution would not have tolerated such an abundant storage of a protein if functioning as an intracellular chaperone were its sole duty, as Csermely and colleagues have long argued [7]. We speculate that a smaller portion of Hsp90α is required to work with Hsp90β for the intracellular duty of chaperones, such as stabilization of HIF-1α [51], whereas the vast majority of eHsp90α is stockpiled for supply to tolerate environmental insults, such as tissue injuries, that take place all the time. The second possible answer is that eHsp90α provides unique properties that are absent from conventional extracellular factors such as cytokines, growth factors, or ECMs. Li’s group showed that topical recombinant eHsp90α protein promoted normal wound healing far more effectively than the (only) FDA-approved growth factor therapy (Raranex^TM^, PDGF-BB). Their study showed that eHsp90α overrides the inhibitory effect of TGFβ family cytokines, which are abundantly present in fresh wounds. To the best of our knowledge, eHsp90α is the first molecule with this unprecedented property [46]. Third, an effective wound-healing agent is one that must recruit all three types of skin cells (epidermal, dermal, and endothelial) to close the wound. However, all growth factors show selectively targeted cell type(s). This limitation has made any single growth factor therapy less effective in the multi-cell process of wound healing. PDGF-BB only acts on dermal fibroblasts, but not epidermal keratinocytes and dermal microvascular endothelial cells, as the latter do not express either PDGFRα or PDGFRβ [46]. These findings may explain why Raranex has shown limited efficacy in clinic, even with several thousand times higher concentration of PDGF-BB (100 μg/g gel) than found in human circulation (0–15 ng/mL). In contrast, eHsp90α acts as a common pro-motility factor for all three types of skin cells involved in wound healing and shows a far stronger effect than PDGF-BB in wound healing [46,65,78]. For similar reasons, eHsp90α may also have an advantage over conventional extracellular factors in cancer invasion. For instance, Hanahan and Weinberg in their heavily cited review on cancer pointed out that one of the most recognized tumour-suppressing effects comes from the anti-growth signal by TGFβ [127]. To sabotage the inhibitory effect of TGFβ, only a small number of tumours choose to mutate either the type II (TβRII) or type I (TβRI) TGFβ receptor or their downstream effector, Smad4, which forms a complex with activated Smad2/3 to regulate gene expression. How the rest of human tumours bypass the TGFβ’s inhibitory signals has never been discussed. We argue that these tumours secrete eHsp90α to override TGFβ inhibition.

## 11. Conclusions and Perspective

It has been the second decade since the official recognition of eHsp90α as a new research branch of Hsp90 in 2008. Since then, all-round progress, including mechanisms of secretion and action, biological function, therapeutic epitope identification, preclinical evaluation, and clinical relevance of eHsp90α, has been reported around the globe. If we have to provide a single outstanding take-home message to the readers, it would undoubtably be the exciting consensus that eHsp90α is not required for homeostasis but remains an essential player under pathological conditions and crisis. Therapeutically targeting eHsp90α in blood circulation represents a particularly exciting modality due to its ease-of-access, safety, and likely increased efficacy compared to targeting intracellular or nuclear Hsp90. For the next decade, the central challenge is to prove the clinical relevance of eHsp90α, such as in tissue injury, fibrosis, and tumorigenesis, and to concurrently establish the druggable window for targeting eHsp90α in human disorders for therapeutic development.

## Figures and Tables

**Figure 1 biomolecules-12-00911-f001:**
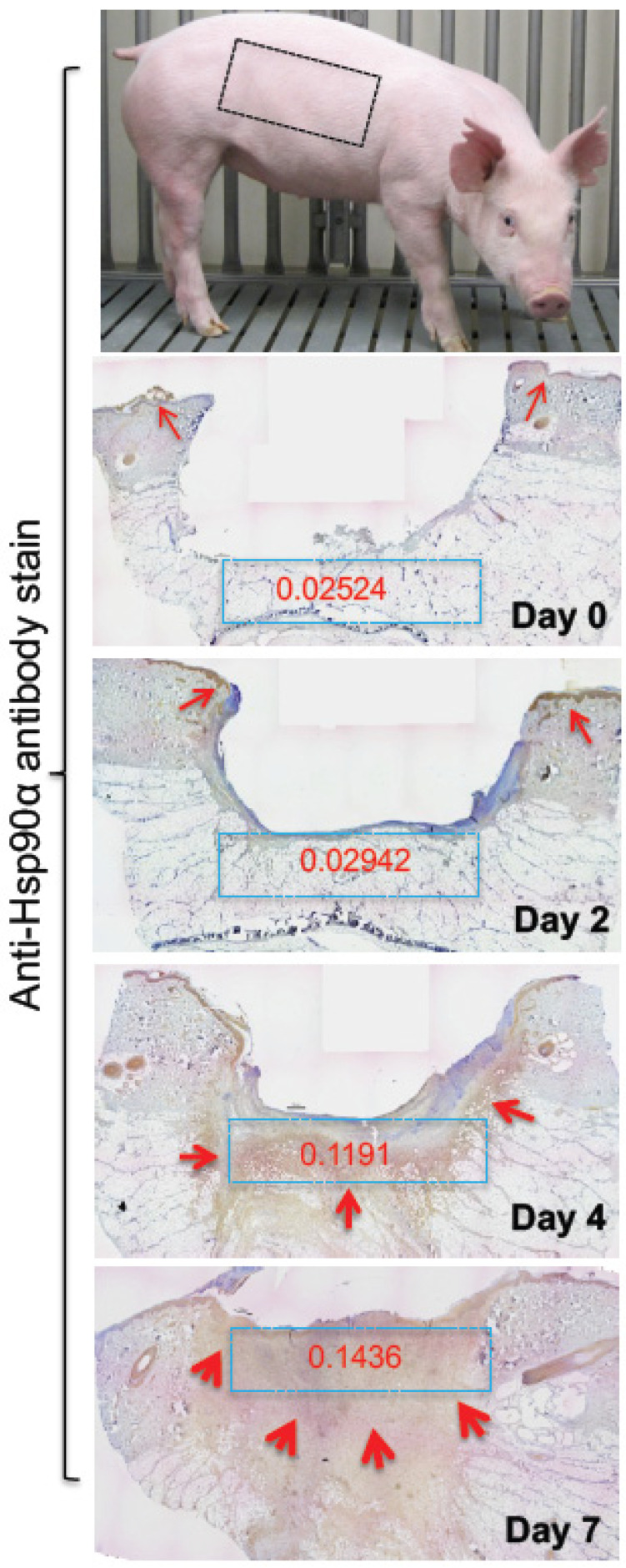
Tissue damage induces massive deposition of eHsp90α into the wound bed. Pig skin is biologically closest to the human skin. 1.5 cm × 1.5 cm full-thickness excision wounds were created in the indicated area of pig torso. Full wedge (2 cm) biopsies cross the wound were made on the indicated days and immediately frozen on dry ice. Sections of the biopsies were stained with an anti-Hsp90α antibody. The red arrows point out the locations of the specific antibody staining (brown). Quantitation of the staining in blue boxes was done using Gabriel Landini’s “color deconvolution” and ImageJ analysis. The intensity readings were converted to Optical Density (OD) (The image was taken from reference [55] with permission).

**Figure 2 biomolecules-12-00911-f002:**
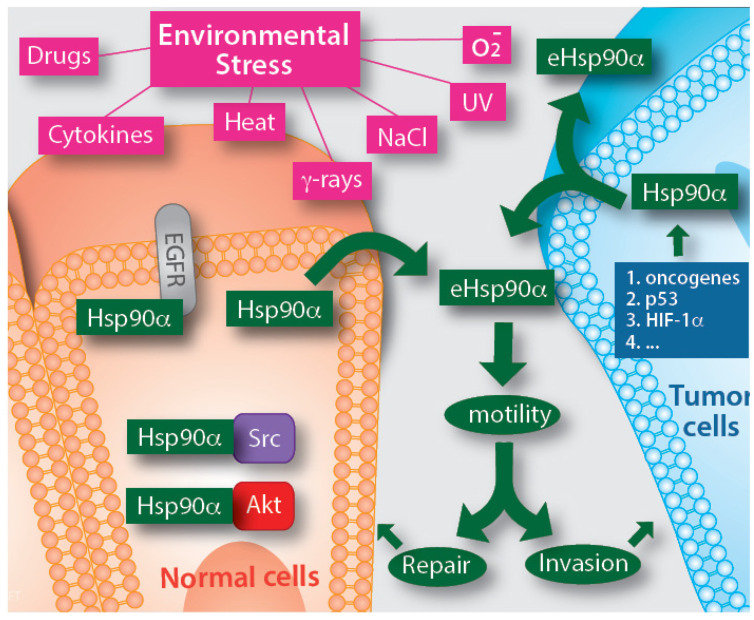
Secretion of eHsp90α by normal cells under medically defined stress and by tumour cells driven by oncogenic signals. Almost all kinds of medically defined stresses have been shown to trigger eHsp90α secretion in a wide variety of cell types. Tumours have either constitutively activated oncogenes or mutant tumour suppressor genes that each triggers eHsp90α secretion even in the absence of environmental stress cues. The mechanisms by which the stress and oncogenic signals cause Hsp90 secretion remain largely unstudied, in which exosome-mediated secretion of Hsp90α only accounts for 10% of the total secreted Hsp90α in both normal and tumour cells. The reported optimal working concentration for the full-length eHsp90α protein was around 3–10 μg/mL.

**Figure 3 biomolecules-12-00911-f003:**
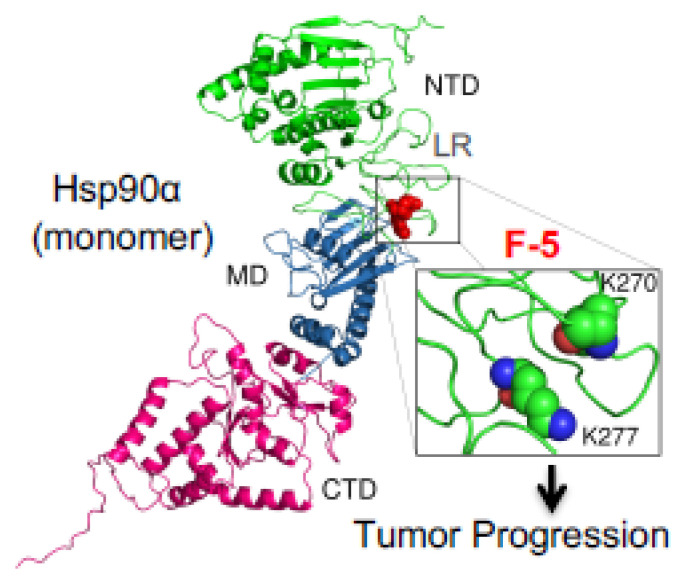
The F-5 fragment is located at the surface of the Hsp90α protein. Based on the previously evaluated crystal structure of a monomer Hsp90α protein with the NTD (green), MD (blue) and CTD (red) domains, the F-5 fragment containing Lysine-270 and lysine-277 is located in the unstructured linker region (LR) between the NTD and MD domains. Inhibitors such as monoclonal antibodies, targeting the dual lysine residues (in enlarged box), are potential anti-tumour therapeutics.

**Figure 4 biomolecules-12-00911-f004:**
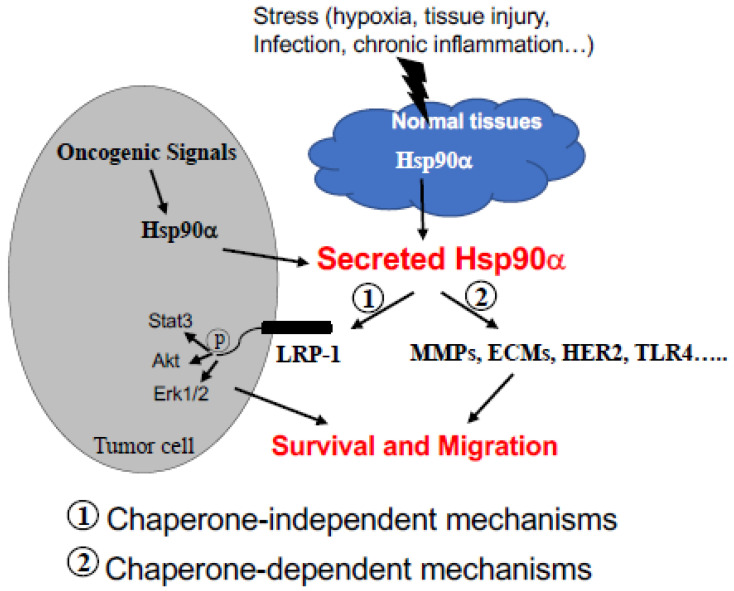
Two proposed mechanisms of action by eHsp90α. eHsp90α acts via an ATPase-dependent or ATPase-independent mechanism, which is determined by different binding partners, as shown. It is possible that the two mechanisms take place in parallel and work synergistically to achieve the ultimate goal under pathophysiological conditions.

**Figure 5 biomolecules-12-00911-f005:**
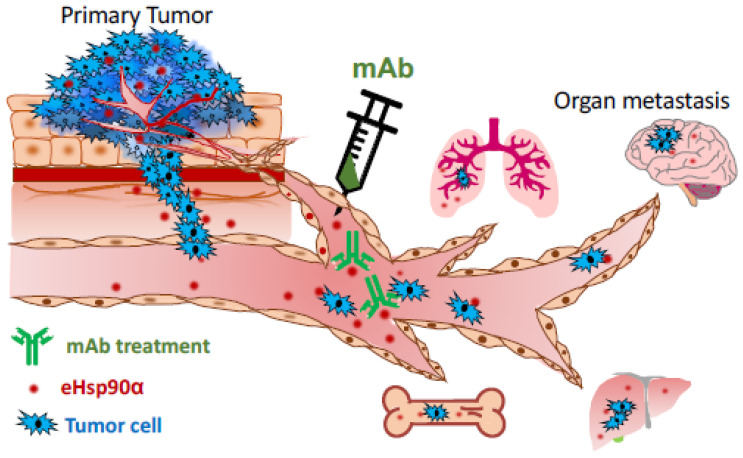
Plasma eHsp90α as a potential target for therapeutics to block tumour metastasis. Findings of the clinical studies shown in Table 1 have raised an exciting possibility that monoclonal antibody therapeutics against plasma eHsp90α block tumour metastasis. Since plasma eHsp90α is low and unessential for homeostasis, targeting plasma eHsp90α in cancer patients may prove to be safer and more effective than targeting the intracellular Hsp90α and Hsp90β.

**Figure 6 biomolecules-12-00911-f006:**
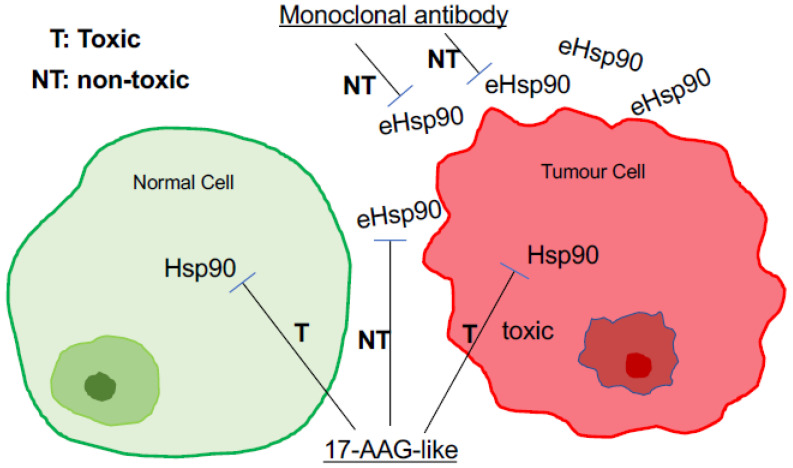
A major difference between targeting eHsp90α and targeting intracellular Hsp90 in cancer. Cytotoxicity and lack of a clear therapeutic window under tolerable dosages have been the major hurdles for ATP-binding inhibitors of Hsp90, especially Hsp90β, in cancer clinical trials. In contrast, selectively targeting eHsp90α with membrane impermeable drug candidates has immerged as a new therapeutic strategy in cancer and beyond.

**Table 1 biomolecules-12-00911-t001:** Summary of clinical studies on plasma eHsp90 in blood circulation *.

Cancer Type	# of Patients	Plasma eHsp90α	# of Healthy Humans	Plasma eHsp90α	Refs.
Mix of liver, lung, breast, colorectal, stomach, pancreatic, esophagus cancer, and lymphoma.	300	IQR 87.01–235.5 Median 157.80 (ng/mL)	132	IQR 22.87–44.46 Median 31.19 (ng/mL)	[91]
Colon (CRC)	635	51.4 (33.8, 80.3) ng/mL	295	43.7 (34.3, 54.8) ng/mL	[92]
Mix of Breast & Other cancers	85	>50 (ng/mL)	16	50.00 (ng/mL)	[56]
Liver	782	IQR 96.7–246.8 Median 159.9 (ng/mL)	572	IQR 21.1–42.2 Median 30 (ng/mL)	[93]
Lung	1046	Ave. 220.46 (ng/mL)	592	Ave. 48.0 (ng/mL)	[94]
Colon (CRC)	77	135 ± 101.94 (ng/mL)	76	44 ± 15.35 (ng/mL)	[95]
Melanoma	98	Median. 49.76 (ng/mL)	43	Median 25.7(ng/mL)	[96]
AML	82	Ave. 295 (ng/mL)	20	Ave. 12.1 (ng/mL)	[97]
Pancreas	20	0.57 ± 0.23 (mg/mL)	10	0.18 ± 0.05 (mg/mL)	[98]
Pancreatic ductal adenocarcinoma	114	1 ± 0.86 (mg/mL)	10	0.18 ± 0.05 (mg/mL)	[98]
Hepatocellular carcinoma	76	274 ± 20.3 (μg/mL)	14	186 ± 18.3 (μg/mL)	[99]
Hepatocellular carcinoma	659	144 ± 4.98 (ng/mL)	230	46 ± 1.11 (ng/mL)	[100]
Esophageal squamous cell carcinoma	193	≥82.06 (ng/mL)			[101]
Esophageal squamous cell carcinoma	93	Ave. 85 (ng/mL)	0	0	[102]
Cervical cancer	220	80.6–212.8 (ng/mL)	75	48.6–89.6 (ng/mL)	[103]
Prostate cancer	18	Median 50.7 (25.5–378.1) (ng/mL)	13	Median 27.6 (13.9–46.5) (ng/mL)	[104]
Childhood acute lymphoblastic leukemia	21	1.22–23.85 (ng/mL)	No exact number	3.16–33.58 (ng/mL)	[105]
Gastric cancer	976	Median 64.3 (ng/mL)	100	45.16 (ng/mL)	[106]
Lung cancer	560	97.64 ± 103.36 (ng/mL)	78	38.44 ± 15.4 (ng/mL)	[107]
Mix of Breast, Liver, Lung, Colon, Esophageal, Gastric and Colorectal	370	57.97–294.63 (ng/mL)	Reference range	0~82.06 (ng/mL)	[108]
Non-small-cell lung cancer	60 Pre-chemotherapy	0.29–0.93 (ng/mL)	60 After 4-cycles of chemotherapy	0.12–0.24 (ng/mL)	[109]
Malignant melanoma	60	70.8–140.77 (ng/mL)	60	42.56–61.42 (ng/mL)	[110]
Nasopharyngeal carcinoma	196	212 ± 144.32 (ng/mL)	106	35 ± 17.47 (ng/mL)	[111]
**Non-cancer diseases**					
Crohn’s disease	53	6.4~55.1			[112]
Psoriasis	80	100 ± 193.66 (AU/mL)	80	63 ± 49.71 (AU/mL)	[113]
Chronic glomerulonephritis	32	33.31–77.25 (ng/mL)	10	22.32	[114]
Amyotrophic lateral sclerosis	58	17.02 ± 10.55	85	12.7 ± 9.23	[115]
Overweight and obese children with Nonalcoholic fatty liver disease	26	3.59–119.85 (ng/mL)	Overweight & obese children without Nonalcoholic fatty liver disease	0–105.4 (ng/mL)	[116]
Chronic glomerulonephritis with nephrotic syndrome	21	33.31–77.25 (ng/mL)	10	Approx. 25–30 (ng/mL)	[114]
Systemic sclerosis	92	9.6–17.9 (ng/mL)	92	7.7–12.4 (ng/mL)	[117]
Diabetic lower extremity arterial disease (DLEAD)	46	Ave. 263.88 (pg/mL)			[11]
Idiopathic pulmonary fibrosis (IPF)	31	Ave. 60 (ng/mL)	9	Ave. 35 (ng/mL)	[118]

* Note: The reported original data on plasma Hsp90 from patients varied dramatically from pg/mL to mg/mL, while the reasons remain unclear. Two presentations, “range” and “average”, by the original studies were adopted here. Nonetheless, higher plasma Hsp90 levels in patients’ blood are evident. IQR: Interquartile range (IQR).

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
