# Peer review of "Extracellular Heat Shock Protein-90 (eHsp90): Everything You Need to Know"

_biomolecules, 2022, doi:10.3390/biom12070911_

Round 1

Reviewer 1 Report

Drs Jay, Luo and Li belong to the list of 'founders' of the modern extracellular Hsp90 field. This is an excellent and well written review, provides a very informative historic context of the eHsp90 field and how it developed over the years. Throughout the review there are critical open-minded discussions that provide many different perspectives and opinions. The authors elegantly cover eHsp90 biological functions, mechanisms, relevance to diseases and future directions with regards to targeting eHsp90. The authors rightly state gaps in the field that need to be studied in the next decade. I foresee this review will be highly cited.

Figures are appropriate and informative.

MInor comments:

1. abstract:  'Doe' should be 'Does'

2. Line 131 - add Eustace reference. Elaborate more on the fact that in this study it was cancer cells other than HT1080 that were tested and showed similar outcome.

3. Line 168 - delete 'identify'

4. Lines 208 - 210 - Not sure if this sentence is supposed to be in this section of the review.

5. Line 247 - MMP-2 change to MMP2

6. Lines 334 - 343 - some fonts sizes need to be checked out.

7.  Are the references formatted twice? 

Author Response

Reviewer #1

Drs Jay, Luo and Li belong to the list of 'founders' of the modern extracellular Hsp90 field. This is an excellent and well written review, provides a very informative historic context of the eHsp90 field and how it developed over the years. Throughout the review there are critical open-minded discussions that provide many different perspectives and opinions. The authors elegantly cover eHsp90 biological functions, mechanisms, relevance to diseases and future directions with regards to targeting eHsp90. The authors rightly state gaps in the field that need to be studied in the next decade. I foresee this review will be highly cited.

Figures are appropriate and informative.

Minor comments:

Question 1: abstract:  'Doe' should be 'Does'

Answer 1: We have made the change in Abstract.

Question 2: Line 131 - add Eustace reference. Elaborate more on the fact that in this study it was cancer cells other than HT1080 that were tested and showed similar outcome.

Answer 2: Ref. 20 is the paper by Eustace. HT1080 is the human cancer (fibrosarcoma) cell line used in the study by Eustace and colleagues in their NCB paper in 2004. Nonetheless, we have made it more clear in the text.

Question 3: Line 168 - delete 'identify'

Answer 3: Yes, deleted.

Question 4: Lines 208 - 210 - Not sure if this sentence is supposed to be in this section of the review.

Answer 4: Agreed! We have deleted this sentence from the section of “mechanism” (BTW, it was hard to satisfy each personal view of three independent investigators/authors).

Question 5: Line 247 - MMP-2 change to MMP2

Answer 5: Has changed.

Question 6: Lines 334 - 343 - some fonts sizes need to be checked out.

Answer 6: Yes, corrected, thanks!

Question 7: Are the references formatted twice? 

Answer 7: The reference was in a little mess, due to additions. We have now re-checked the references on by one to match their numbers in the text.

Reviewer 2 Report

Review of Jay, Luo, and Li Biomolecules review of eHsp90

This review of eHsp90 covers its various roles and mechanisms of action in detail. However, it is very poorly and awkwardly written and will require significant and careful editing by someone whose first language is English. Grammatical errors, incomplete sentences, etc. (in both main text and figure legend) are too numerous to specifically identify but occur throughout the manuscript.

Other concerns/comments:

1.     ‘hr’Hsp90 on line 106 is not defined until later in the manuscript. Please define hrHsp90 here as human recombinant Hsp90.

2.     Line 148: should ‘secretion’ appear after Hsp90a?

3.     In the sentence beginning on line 256, it would be helpful to state that the Hsp90-E47 mutants bind ATP but cannot hydrolyze it, while the D93 mutant does not bind ATP. Thus, the E47 mutant achieves a closed conformation (N-domain dimerization) in contrast to the D93 mutant which remains primarily in an open, N-domains undimerized conformation.

4.     I would suggest adding a figure showing a simple schematic of Hsp90 monomer structure, in which the N-domain, charged linker region, middle domain and C-terminal domain are identified, as is the location of the F-5 peptide. I think this would be very beneficial to the reader.

5.     The beginning of section 7.1 (starting on line 299) is redundant with an earlier section starting on line 185.

6.     I would suggest including another Table listing papers that suggest the ATP dependence of eHsp90 function and those that do not. In the same table, papers suggesting that F-5 peptide is sufficient for eHsp90 function should also be listed. In reading the text, I get quite confused trying to keep straight the importance of various parts of eHsp90, dependence on ATP binding or ATPase activity etc., as the authors repeatedly go back and forth on these contrasting requirements throughout the review.

7.     Re Table 1: what does # of control mean? Should this be % of control? In either case, how is this number determined and what does it mean to have a number in this column that is > 0?

8.     On the second page of Table 1, I would group all the non-cancer indications together at the bottom of the table.

Author Response

Reviewer # 2:

Review of Jay, Luo, and Li Biomolecules review of eHsp90

This review of eHsp90 covers its various roles and mechanisms of action in detail. However, it is very poorly and awkwardly written and will require significant and careful editing by someone whose first language is English. Grammatical errors, incomplete sentences, etc. (in both main text and figure legend) are too numerous to specifically identify but occur throughout the manuscript.

Other concerns/comments:

Question 1: ‘hr’Hsp90 on line 106 is not defined until later in the manuscript. Please define hrHsp90 here as human recombinant Hsp90.

Answer 1: Yes, we have now defined hrHsp90 on line 106, where it first appeared.

Question 2:   Line 148: should ‘secretion’ appear after Hsp90a?

Answer 2: Yes, now changed to “Hsp90a secretion”

Question 3:  In the sentence beginning on line 256, it would be helpful to state that the Hsp90-E47 mutants bind ATP but cannot hydrolyze it, while the D93 mutant does not bind ATP. Thus, the E47 mutant achieves a closed conformation (N-domain dimerization) in contrast to the D93 mutant which remains primarily in an open, N-domains undimerized conformation.

Answer 3: Considering the specific comment of the reviewer on the Hsp90 mutants and the focus of this article on “extracellular Hsp90”, we do not feel the necessity to go into the conformational details on these mutations, rather we have added: “Hsp90a-E47A (~50% ATPase activity), Hsp90a-E47D (ATPase-defect), and Hsp90a-D93N (ATPase-defect)”.  

Question 4: I would suggest adding a figure showing a simple schematic of Hsp90 monomer structure, in which the N-domain, charged linker region, middle domain and C-terminal domain are identified, as is the location of the F-5 peptide. I think this would be very beneficial to the reader.

Answer 4: Agreed! Please see the new Figure 3.

Question 5:  The beginning of section 7.1 (starting on line 299) is redundant with an earlier section starting on line 185.

Answer 5: Yes, we have made the changes.

Question 6: I would suggest including another Table listing papers that suggest the ATP dependence of eHsp90 function and those that do not. In the same table, papers suggesting that F-5 peptide is sufficient for eHsp90 function should also be listed. In reading the text, I get quite confused trying to keep straight the importance of various parts of eHsp90, dependence on ATP binding or ATPase activity etc., as the authors repeatedly go back and forth on these contrasting requirements throughout the review.

Answer 6: This point is sensitive. There have been two major parallel mechanisms of action proposed for eHsp90a. The central debate is whether eHsp90a still acts as an ATP-dependent chaperone outside the cell or alternatively acts as a previously unrecognized signalling molecule no longer dependent on ATP hydrolysis. In agreement with Dr. Dimitra Bourboulia, who also contributes a review article to this special issue on eHsp90a as a ATPase-dependent chaperone, I would not discuss the “ATP-dependent chaperone mechanism” of eHsp90a in any detail. She will do it in her article.

Question 7:  Re Table 1: what does # of control mean? Should this be % of control? In either case, how is this number determined and what does it mean to have a number in this column that is > 0?

Answer 7: “Control” means “non-cancer humans”. We have changed “control” to “healthy humans”. We only reported the original findings from each of the clinical studies (Some of them had missing information, which was out of our control). Nonetheless, we have gone back to have re-checked the data and modified our presentations. We have made necessary changes and added explanations at the bottom of the Table.  

Question 8: On the second page of Table 1, I would group all the non-cancer indications together at the bottom of the table.

Answer 8: Great idea! We have done it now.

Reviewer 3 Report

In the present manuscript Jay, Luo, and Li expertly review the extracellular Hsp90 field. Their review provides the historical basis for the discovery and recognition of eHsp90. They subsequently walk the reader through the studies describing how this protein was found to exist extracellularly and why cells secrete it. The functions of eHsp90 in wound healing, fibrosis, and tumor invasion and migration are detailed as well as how this has been targeted in preclinical models and measured in clinical applications. The authors then highlight eHsp90 as a drug target and place this in the context of the tribulations of trials of small molecule Hsp90 inhibitors designed to target the intracellular form. 

The authors quite clearly have a unique perspective on this relatively new field and use this to guide the reader through what has been found about eHsp90 and to lay the foundation for what work is on the horizon and is needed. The present manuscript was enjoyable to read and no major corrections are suggested by this reviewer. The authors and journal should ensure high quality images for the figures (Figure 3 and Table 1 were quite fuzzy) and consistency of the text (for example, different font and size in lines 339-343) as well as some minor copyediting. 

Author Response

Reviewer #3:

In the present manuscript Jay, Luo, and Li expertly review the extracellular Hsp90 field. Their review provides the historical basis for the discovery and recognition of eHsp90. They subsequently walk the reader through the studies describing how this protein was found to exist extracellularly and why cells secrete it. The functions of eHsp90 in wound healing, fibrosis, and tumor invasion and migration are detailed as well as how this has been targeted in preclinical models and measured in clinical applications. The authors then highlight eHsp90 as a drug target and place this in the context of the tribulations of trials of small molecule Hsp90 inhibitors designed to target the intracellular form. 

Question: The authors quite clearly have a unique perspective on this relatively new field and use this to guide the reader through what has been found about eHsp90 and to lay the foundation for what work is on the horizon and is needed. The present manuscript was enjoyable to read and no major corrections are suggested by this reviewer. The authors and journal should ensure high quality images for the figures (Figure 3 and Table 1 were quite fuzzy) and consistency of the text (for example, different font and size in lines 339-343) as well as some minor copyediting. 

Answer: Great suggestions! We have modified Figure 3 (especially the colors). Table never looks clear on PPT, so that we have remade the table in word file and attached it the MS. This revision has allowed us to go back to re-check every word, every image and every reference, and made them right, for which we feel grateful.

Round 2

Reviewer 2 Report

Revised version is acceptable without additional modifications.